# Hypoxic pregnancy promotes fibrosis and increases stress metabolites in the ovine fetal liver

Molly M. McGuckin[1] , Dong Wang[1], Jasmine Ortiz[1], Evgenia Dobrinskikh[1], Wen Tong[2,3], Kimberley J. Botting-Lawford[2], Youguo Niu[2,3] , Dino A. Giussani[2,3,4] and Stephanie R. Wesolowski[1]

[1] *Department of Pediatrics, University of Colorado School of Medicine, Aurora, CO, USA*
[2] *Department of Physiology, Development and Neuroscience, University of Cambridge, Cambridge, UK*
[3] *The Loke Centre for Trophoblast Research, University of Cambridge, Cambridge, UK*
[4] *The Cambridge Reproduction Interdisciplinary Research Centre, University of Cambridge, Cambridge, UK*

Handling Editors: Laura Bennet & Kyle McCommis

The peer review history is available in the Supporting information section of this article (https://doi.org/10.1113/JP288724#support-information-section).

**Abstract figure legend** During hypoxic pregnancy, late gestation fetuses had increased periportal hepatic collagen deposition, an indicator of hepatic injury, with a unique profile of stress metabolites and adaptations in central carbon metabolism. This included increased hepatotoxic 5-oxyproline and sphingosphine-1-phosphate, which promote the

The Journal of Physiology

chemotaxis of collagen-producing cells. Hypoxic fetal livers also had decreased glutathione (GSH) building blocks glycine (Gly), serine (Ser) and glutamyl-cysteine (GluCys) and downregulation of glutathione synthase (GSS). The abundance and activity of lactate dehydrogenase enzyme (LDHA) was increased, which converts $\alpha$-ketoglutarate to 2-hydroxyglutarate, a metabolite implicated in liver injury and inflammation via epigenetic mechanisms. Furthermore, there was predicted regulation of lysine demethylase 5A (KDM5A), which is inhibited by $S$-adenosyl-L-methionine (SAM), a decreased metabolite in hypoxic livers. Combined, this hepatic profile supports a pro-fibrogenic *milieu* that provides new insight into how pregnancy complicated by hypoxia initiates hepatocellular damage and disrupts metabolism in the fetus.

**Abstract** Fetal chronic hypoxia is a common pregnancy complication associated with fetal growth restriction. Growth-restricted offspring have a higher risk for liver metabolic disease. Our objective was to better understand how chronic hypoxia impacts the developing fetal liver. We hypothesized that hypoxia promotes hepatocellular injury, shifts nutrient metabolism, and activates energetic and oxidative stress in the fetal liver. We used an ovine model of chronic hypoxia where pregnant ewes were housed under normoxic (CON) or hypoxic (HOX) conditions for 30 days in late gestation. Fetal liver was obtained, histologically analysed and profiled using bulk-RNA sequencing and metabolomics. Nutrient and oxidative stress signalling pathways were also measured. HOX fetuses had greater hepatic periportal collagen deposition. Metabolomics and transcriptomics predicted disruptions in central carbon metabolism, mitochondrial dysfunction and decreased oxidative phosphorylation. In support, we found potentiation of the gluconeogenic pathway and increased lactate production, pyruvate oxidation and AMPK activation. By contrast to the predicted effects, hypoxic livers maintained mitochondrial oxidation and antioxidant capacity. Interestingly, acylcarnitines were increased, yet hepatic triglyceride content was similar. Although there was little activation of oxidative stress markers, such as lipid peroxidation or oxidized glutathione, we uncovered a unique profile of liver stress-related metabolites in association with periportal collagen. Thus, hypoxic pregnancy increased fetal hepatic collagen deposition, indicating liver injury, in association with a unique profile of liver stress metabolites and adaptations in central carbon metabolism. These results provide new insight into how chronic fetal hypoxia may initiate fibrotic and metabolic liver disease risk in offspring of adverse pregnancy.

(Received 12 February 2025; accepted after revision 3 April 2025; first published online 30 April 2025)

**Corresponding authors** S. R. Wesolowski: Perinatal Research Center, University of Colorado School of Medicine, Mail Stop F441, Aurora, CO 80045, USA. Email: stephanie.wesolowski@cuanschutz.edu

D. A. Giussani: Department of Physiology, Development, and Neuroscience, University of Cambridge, Downing Street, CambridgeCB2 3EG, United Kingdom. Email: dag26@cam.ac.uk

**Key points**

- Chronic exposure to hypoxic pregnancy increased fetal hepatic collagen deposition, indicating hepatocellular injury.
- Hypoxic fetal livers had a unique profile of stress metabolites and adaptations in central carbon metabolism.
- This provides new insight into how hypoxia, a common pregnancy complication associated with fetal growth restriction, may initiate fibrotic and metabolic liver disease risk.

**Molly McGuckin** is a postdoctoral research fellow at the University of Colorado School of Medicine, specializing in fetal physiology and liver metabolism. Her research examines how intrauterine conditions, such as growth restriction and hypoxic pregnancy, impact fetal liver metabolism. She earned her PhD in Animal Science from Cornell University, where her work on metabolic signalling in dairy cows laid the foundation for her current research. She is a recipient of the National Institutes of Health F32 postdoctoral fellowship and aims to understand the *in utero* origins of metabolic diseases.

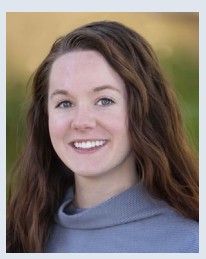

## Introduction

Fetal chronic hypoxia is a feature of high-altitude pregnancy and sea level pregnancy complicated by placental insufficiency, pre-eclampsia and other conditions that increase placental vascular resistance (Ducsay et al., 2018; Giussani, 2016). Fetal growth restriction (FGR) is a frequent consequence of adverse pregnancy, and offspring born with FGR have an increased life-long risk of developing cardiometabolic disease (Giussani, 2021; Jornayvaz et al., 2004; Mericq et al., 2017; Nobili et al., 2008) and liver disease associated with fibrosis, such as total parenteral nutrition-induced neonatal cholestasis (Champion et al., 2012; Feldman & Sokol, 2019). Individuals born with FGR also have dysregulated hepatic glucose production as neonates (Cowett et al., 1983; Kalhan et al., 1986), higher rates of hepatic insulin resistance (Brons et al., 2008, 2022; Dufour & Petersen, 2011; Stefan et al., 2004) and steatotic liver disease with higher hepatic fibrosis scores as children and adults (Newton et al., 2017; Nobili et al., 2008). However, it remains unclear how fetal chronic hypoxia during FGR initiates metabolic disease risk in the fetal liver.

As in the adult, the fetal liver is a metabolic hub, co-ordinating the metabolism of glucose, lactate, amino acids and lipids based on the incoming supply with the metabolic demands of the liver and extrahepatic tissues. Notably, the fetal liver receives the first pass at nutrient- and oxygen-rich umbilical venous blood from the placenta. This makes the fetal liver vulnerable during hypoxic conditions when oxygen supply is decreased and the nutrient *milieu* is altered (Jones et al., 2021). In a sheep model of placental insufficiency-induced FGR, with chronic and progressive reductions in oxygen and nutrient transport to the fetus, these fetuses show an early activation of hepatic gluconeogenic gene expression and glucose production (Brown et al., 2024; Thorn et al., 2013). To tease apart the effect of reduced oxygen supply, prior studies showed that experimental hypoxia for 10 days in late gestation fetal sheep potentiated hepatic gluconeogenic gene expression (Jones et al., 2019). In guinea-pigs, gestational hypoxia also increases lipid peroxidation and glutathione content in the fetal liver (Al-Hasan et al., 2013; Hashimoto et al., 2012; Oh et al., 2008) and produces liver tissue damage in postnatal offspring (Sarr et al., 2016). Furthermore, FGR offspring, across models, are at increased risk for developing mitochondrial dysfunction (Pendleton et al., 2021). However, there are gaps in our understanding of how exposure to chronic hypoxia *in utero* may trigger metabolic adaptations in the fetal liver that contribute to liver metabolic disease risk throughout the lifespan of exposed offspring.

The objective of this study was to delineate how chronic hypoxia disrupts metabolism and development in the fetal liver *in utero*. We hypothesized that hypoxia promotes hepatocellular injury, shifts nutrient metabolism and activates energetic and oxidative stress signalling pathways in the fetal liver. This was tested using an established ovine model of isobaric hypoxic pregnancy for 30 days during late gestation (Allison et al., 2016, 2020; Botting et al., 2020; Brain et al., 2015, 2019; Tong et al., 2022), providing a longer duration over previous 10 day hypoxia models (Jones et al., 2021, 2022). Furthermore, this longer-term hypoxic pregnancy model isolates the contribution of hypoxia, which is confounded with reduced nutrient supply in placental insufficiency models (Kyllo et al., 2023). To investigate the effect of hypoxic pregnancy, we utilized comprehensive fetal liver histopathology analysis, unbiased metabolomic and transcriptomic profiling and confirmation experiments focused on nutrient- and oxidative stress signalling mechanisms. We also determined the relationship between putative causative factors that may initiate fetal hepatocellular injury. Results from these integrative approaches advance our understanding of the impact of chronic hypoxia on the developing fetal liver.

## Methods

### Ethical approval

Animal procedures were conducted at the Barcroft Centre of the University of Cambridge and approved by the Ethical Review Board of the University of Cambridge (Home Office Project Licence PP6755721). Research was conducted and reported in adherence to the Animal Research: Reporting of In Vivo Experiments (ARRIVE) guidelines (Percie du Sert et al., 2020).

### Ovine model of hypoxic pregnancy

The hypoxic pregnancy model and animals used in the present study have been previously reported (Allison et al., 2016, 2020; Botting et al., 2020; Brain et al., 2015, 2019; McGillick et al., 2017; Tong et al., 2022), yet our study is the first to measure outcomes in the fetal liver. In brief, pregnant Welsh Mountain ewes carrying singletons were studied. Ewes were fed a maintenance diet daily and had free access to water. Feed intake and medical records were maintained daily, along with maternal body weight measurements. Beginning at 105 days gestational age (dGA) (term is ~147 days), ewes were randomly assigned to hypoxic (HOX; $n = 7$) or normoxic (CON; $n = 9$) pregnancy, after which the study was performed unblinded to maintain experimental conditions. Hypoxic pregnancy in late gestation was induced by housing ewes in bespoke isobaric hypoxic chambers to maintain the HOX group at 10% inspired $O_2$ or atmospheric $O_2$ for

CON (Allison et al., 2016, 2020; Botting et al., 2020; Tong et al., 2022).

## Blood analysis and tissue collection

At 138 dGA, hypoxic ewes were removed from the chambers and maintained at 10% $O_2$ in a transport cart until the time of death and were humanely killed by intravenous overdose of sodium pentobarbitone (0.4 mL kg$^{-1}$; Pentoject; Animal Ltd, York, UK). Maternal and fetal arterial blood samples were collected to measure haemoglobin as previously reported (Tong et al., 2022). Fetuses were delivered by hysterectomy and organs were weighed. Fetal liver tissue was isolated and fixed in 4% paraformaldehyde for histology or flash frozen in liquid nitrogen and stored at −80°C. End-points were carefully selected to prevent confounding factors such as changes in glucose metabolism and cortisol spikes that occur closer to term (Fowden et al., 1996).

## Histological analysis

Formalin-fixed and paraffin-embedded liver tissue was used for histological analysis, as previously described (Nash et al., 2023). Tissue sections were cut to 5 μм thickness and placed on glass slides. Samples were deparaffinized, hydrated and stained with haematoxylin and eosin (H&E) (VectorLabs, Newark, CA, USDA; no. H-3502) to identify liver morphology or picrosirius red (PSR) (Polysciences, Inc., Warrington, PA, USA; no. 24901) to stain total collagen. Tissue sections were scanned using Aperio CS2 Digital Pathology Slide Scanner (Leica Biosystems, Deer Park, IL, USA) at 40× magnification. Representative 500 μм$^2$ fields of view containing one portal triad (PT) and central vein (CV) region were taken (10–15 fields of view per region) in QuPath, version 0.4.2 (Bankhead et al., 2017) and exported to ImageJ (NIH, Bethesda, MD, USA) to quantify the PSR staining area. The average PSR-stained area was calculated as the percentage of stained pixels relative to total area for all PT and CV regions per animal. One animal was excluded from histological analysis as a result of error in sample preparation.

To quantify fibrillar collagen, we performed second harmonic generation (SHG) imaging on unstained paraffin tissue sections, as previously described (Nash et al., 2021). The SHG signal was acquired using a Zeiss 780 LSM laser-scanning confocal/multiphoton-excitation fluorescence microscope (Carl Zeiss, Oberkochen, Germany). Imaging settings were configured to maximize signal-to-noise ratio and avoid signal saturation. Settings were kept constant for comparative imaging and analysis. Two percent of a two-photon Mai Tai (Spectra-Physics, Milpitas, CA, USA) laser tuned to 800 nm was used for excitation and emission signals corresponding to autofluorescence and SHG. Signals were detected simultaneously with non-descanned detectors and processed using Zeiss Zen 2012 software. Ten to fifteen 500 μm$^2$ fields of view containing one PT or one CV were obtained for each sample. Images were analysed with ImageJ to quantify SHG signal area and intensity, and averages were calculated for all PT and CV regions per animal.

## Targeted metabolomics analysis

Semi-targeted metabolomic profiling was performed in liver tissue samples, as previously described (Brown et al., 2024; Nash et al., 2023). Frozen ground liver tissue (25 mg) samples were extracted in ice-cold lysis buffer 5:3:2 MeOH:MeCN:water via vortexing for 30 min at 4°C followed by centrifugation (18 000 **g** for 10 min at 4°C). Samples were analysed using a Vanquish UHPLC paired with a Q Exactive MS (Thermo Fisher Scientific, Waltman, MA, USA) at the University of Colorado Metabolomics Core (Aurora, CO, USA). Peak intensity values were assigned using MAVEN (Princeton University, Princeton, NJ, USA), as previously described (Brown et al., 2024; Nash et al., 2023). Data were processed using log-transformation and auto-scaling in MetaboAnalyst, version 6.0 (Pang et al., 2024). Multivariate principal component analysis was performed using partial least squares discrimination (PLS-DA) and variable in projection (VIP) scores were based on PLS-DA to identify the most influential metabolites. Select metabolites and ratios were calculated using raw abundance and compared using the Student's *t* test or the Mann–Whitney *U* test, as appropriate.

## Triglyceride assay

Frozen fetal liver was homogenized for extraction of lipids and triglyceride content was measured using Infinity Triglyceride Reagent (Thermo Fisher Scientific; no. TR22421) (McCurdy et al., 2009; Thorn et al., 2014). Triglyceride content was normalized to tissue weight.

## Gene expression

Frozen fetal liver tissue was homogenized in TRIzol (Qiagen, Valencia, CA, USA; no. 15596026), and RNA was extracted using the RNeasy mini spin columns kit (Qiagen). Bulk RNA sequencing was conducted at the University of Anschutz Medical Campus Genomics and Microarray Core (Aurora, CO, USA). Total RNA integrity was measured using the TapeStation 4200 (Agilent, Carpinteria, CA, USA), and RIN values ranged from 6.9 to 8.0. Libraries were constructed using 100 ng of total RNA using the Universal Plus mRNA library prep kit (Tecan, Redwood City CA, USA; no. 0508). PolyA-enriched RNAs

were sequenced at a depth of 80 million paired end reads at $2 \times 150$ bp using the NovaSEQ 600 (Illumina, San Diego, CA, USA). Mean quality score per sample ranged from 35.33 to 35.44 and >90% of reads had a quality score >30. Sequences were mapped to the *Ovis aries* reference genome (ARS-UI_Ramb_v1.0) and *O. aries rambouillet* annotation file (Ensembl) using the STAR computation pipeline (Dobin & Gingeras, 2015). The average number of total reads per sample was 58 164 696 reads, and 66% of input reads were uniquely mapped to the reference genome. Official HUGO gene symbols were confirmed for existing genes and updated for genes with >80% homology to human homolog using BioMart (Ensembl). Gene expression analysis was conducted using R package DEseq2 (Love et al., 2014). Genes were filtered and excluded if raw reads were <250 for more than half the samples leaving 8727 expressed genes (26 478 total genes). False discovery rate (*Q* value) was calculated using the Benjamini–Hochberg procedure, and differentially expressed genes (DEGs) were declared significant when $Q < 0.15$.

### Predicted pathway enrichment

Ingenuity pathway analysis (Qiagen) was used to predict canonical pathways enriched in DEGs using log fold changes and *Q* values. The list of genes that passed expression filtering was used as reference. Pathways were considered enriched when $P < 0.05$ with predicted pathway activation indicated by positive or negative *Z*-score. Upstream regulators were predicted ($P < 0.05$) in the following categories: cytokine, enzyme, growth factor, kinase, peptidase, phosphatase, transcription regulator, translation regulator or transporter. Positive *Z*-scores indicate predicted pathway activation, and negative *Z*-scores indicate predicted inhibition. We used the MetaboAnalyst Pathway Joint Analysis module to predict impacted metabolic pathways ($P < 0.05$), using the DEGs and the 40 metabolites with the highest-ranking VIP scores.

### Western immunoblotting

Liver protein lysates were prepared as previously described (Brown et al., 2024; Jones et al., 2019). Protein concentrations were measured using the Pierce BCA Protein Assay (Thermo Fisher Scientific; no. 23225). Primary antibodies were AMP-activated protein kinase (AMPK; dilution 1:1000; Cell Signaling, Danvers, MA, USA; no. 2532), phosphorylated AMPK[Thr172] (ph-AMPK; dilution 1:1000; Cell Signaling; no. 2531S), mammalian target of rapamycin (mTOR; dilution 1:500; Cell Signaling; no. 4517), phosphorylated mTOR[Ser2448] (ph-mTOR; dilution 1:500; Cell Signaling; no. 2971), eukaryotic translation initiation factor 4E binding protein 1, (4EBP1; dilution 1:1000; Cell Signaling; no. 9452), phosphorylated 4EBP1[Thr39/49] (ph-4EBP1; dilution 1:1000; Cell Signaling; no. 9459), ribosomal protein S6 (S6; dilution 1:1000; Cell Signaling; no. 2217), phosphorylated S6[Ser235/236] (ph-S6; dilution 1:1000; Cell Signaling; no. 2211), glucose-6-phosphatase catalytic subunit 1 (G6PC; dilution 1:500; Abcam, Cambridge, UK; no. ab83690), phosphoenolpyruvate carboxykinase 1 (PCK1; dilution 1:1000; Abcam; no. 70358), pyruvate dehydrogenase (PDH; dilution 1:500; Abcam; no. 110330), phosphorylated PDH[Ser293] (ph-PDH; dilution 1:500; Abcam; no. 92696), lactate dehydrogenase A (LDHA; dilution 1:1000; Cell Signaling; no. 47010), total OXPHOS antibody cocktail (OXPHOS; dilution 1:1000; Abcam; no. ab110413), nuclear factor erythroid 2-related factor 2 (NRF2; dilution 1:1000; Invitrogen, Waltham, MA, USA; no. PA5-27882), superoxide dismutase 2 (SOD2; dilution 1:1000; Millipore, Burlington, MA, USA; no. 06-984) and NADPH quinone dehydrogenase 1 (NQO1; dilution 1:1000; Cell Signaling; no. 62262S). Antibody specificity was verified by a single band at the expected molecular weight and using antibodies previously validated in the sheep (Brown et al., 2024; Jones et al., 2022). Total protein was quantified with Revert 700 Total Protein Stain (LI-COR, Lincoln, NE, USA; no. 926-11010) and used for normalization of target bands. Protein bands were visualized using IRDye 800CW goat anti-Rabbit IgG (LI-COR; no. 926-32211) or IRDye 680RD goat anti-Mouse IgG (LI-COR; no. 926-68070) secondary antibodies. Protein expression was quantified with Image Studio (LI-COR) and expressed relative to total protein. Data are presented relative to the mean of the CON group.

### Enzyme activity

To measure pyruvate dehydrogenase (PDH) activity (Brown et al., 2024; Jones et al., 2021), tissue samples (40 mg) were homogenized in 400 mL of ice-cold PDH assay buffer (Sigma Aldrich, St Louis, MO, USA; no. MAK183). Assays were performed at 37°C using 10 mg of protein. $A_{450}$ was measured every 5 min and the difference $A_{450}$ between 0 and 30 min was calculated to represent the production of NADH resulting from the conversion of pyruvate to acetyl-CoA by PDH enzymatic activity. To measure lactate dehydrogenase (LDH) activity (Brown et al., 2024; Jones et al., 2021), tissue samples (50 mg) were homogenized in 500 mL of ice-cold CelLytic MT Buffer (Sigma-Aldrich; no. C3228). LDH Activity Assay (Abcam; no. ab102526) was used, and reactions were performed with 1 mg of protein for 40 min at 37°C. $A_{450}$ was measured every 5 min and the difference between 10 and 15 min was calculated to represent the production of NADH by the conversion of lactate into pyruvate by LDH enzymatic activity. Protein concentrations were measured

**Table 1. Maternal and fetal characteristics.**

| Variable | CON | HOX | *P* |
|---|---|---|---|
| *N* | 9 | 7 | |
| Male:female | 5:4 | 3:4 | |
| Body weight (kg) | 4.9 ± 0.5 | 3.7 ± 0.4 | <0.001 |
| Liver weight (g) | 106.7 ± 17.3 | 70.2 ± 2.5 | 0.00102 |
| Brain/body weight (g kg$^{-1}$) | 9.8 ± 1.0 | 12.9 ± 1.3 | <0.001 |
| Liver/body weight (g kg$^{-1}$) | 21.8 ± 2.5 | 18.6 ± 2.9 | 0.0349 |
| Haemoglobin (g L$^{-1}$) | 145.4 ± 14.3 | 170.3 ± 15.0 | 0.00793 |
| Maternal bodyweight (kg) | 55.6 ± 6.9 | 50.1 ± 5.3 | 0.0888 |
| Maternal haemoglobin (g L$^{-1}$) | 128.8 ± 12.1 | 159.1 ± 9.7 | <0.001 |

Measurements were taken on the day of fetal necropsy. Values are the mean ± SD in control (CON) or hypoxic (HOX) ewes and fetuses exposed to control (CON) or hypoxic (HOX) pregnancy. P values calculated by Student's *t* test.

(BCA assay), and data were normalized to protein loaded per reaction.

### Thiobarbituric acid-reactive substances

Thiobarbituric acid-reactive substances (TBARS) content was measured colorimetrically (Cayman, Ann Arbor, MI, USA; no. 700 870), as previously described (Brown et al., 2024). Protein samples were used as described for Western blotting. Results are expressed relative to protein content.

### Statistical analysis

Unless otherwise noted, statistical analysis was performed using Prism, version 9.0 (GraphPad Software Inc., San Diego, CA, USA). Collagen deposition was analysed by two-way analysis of variance (ANOVA) with fixed effects of region (PT and CV), treatment (CON and HOX), as well as the interaction, with a random effect to account for repeated measures (i.e. two regions) on samples from the same fetus. Multiple comparisons between groups were made using Tukey's HSD *post hoc* analysis, and different letters were used to indicate variation between groups. Other data were analysed using an unpaired two-tailed Student's *t* test between HOX and CON groups or the Mann–Whitney *U* test when variances were different. Pearson's correlation coefficients were calculated using raw collagen deposition (% area SHG and PSR), blood haemoglobin content and normalized metabolite data. Each animal was considered a replicate, and the average of technical replicates were used in molecular assays. All data were analysed, and no outliers were excluded. Our study was not powered to detect sex differences, but previous studies show no sex differences for most fetal nutrient uptake and liver outcomes in fetal sheep (Brown et al., 2022; Jones et al., 2019, 2022). Data are presented as the mean ± SD.

## Results

### Fetal characteristics

Characteristics of fetuses from normoxic (CON) and hypoxic (HOX) pregnancies are listed in Table 1. Late gestation HOX fetuses, studied at ∼90% of gestation, were asymmetrically growth-restricted compared to CON fetuses, weighing 24.5% less ($P < 0.001$) with increased brain-to-body weight ratio (1.3-fold, $P < 0.001$). Liver weight was 34.2% less ($P < 0.00102$) in HOX compared to CON fetuses. Additionally, blood haemoglobin concentration was 1.3-fold higher ($P = 0.00793$), supporting a physiological response to hypoxia. Additional phenotypic details have been reported on these animals (Allison et al., 2016; Brain et al., 2015; Tong et al., 2022). Maternal characteristics are reported in Table 1.

### Hypoxia increased periportal collagen deposition

We performed histological analysis of fetal livers to determine whether hypoxic pregnancy alters hepatocellular development. We observed no gross histological differences with H&E staining between CON and HOX (Fig. 1*A*). Using PSR staining, a measure of total collagen deposition, we found increased collagen in periportal regions surrounding the PT compared to perivenous regions surrounding the CV in the fetal livers from both groups (region effect, $P < 0.0001$) (Fig. 1*B*). Within PT regions, HOX fetuses had increased PSR staining compared to CON fetuses, with no difference between groups in CV regions (INT effect, $P = 0.0470$) (Fig. 1*B*). Deeper analysis used SHG imaging to quantify fibrillar collagen, a highly sensitive method for visualizing structural proteins in unstained tissue (Nash et al., 2021). SHG signal area and intensity were both higher in PT compared to CV regions (region effect, $P < 0.0001$) (Fig. 1*C* and *D*). Similar to PSR staining, both SHG signal

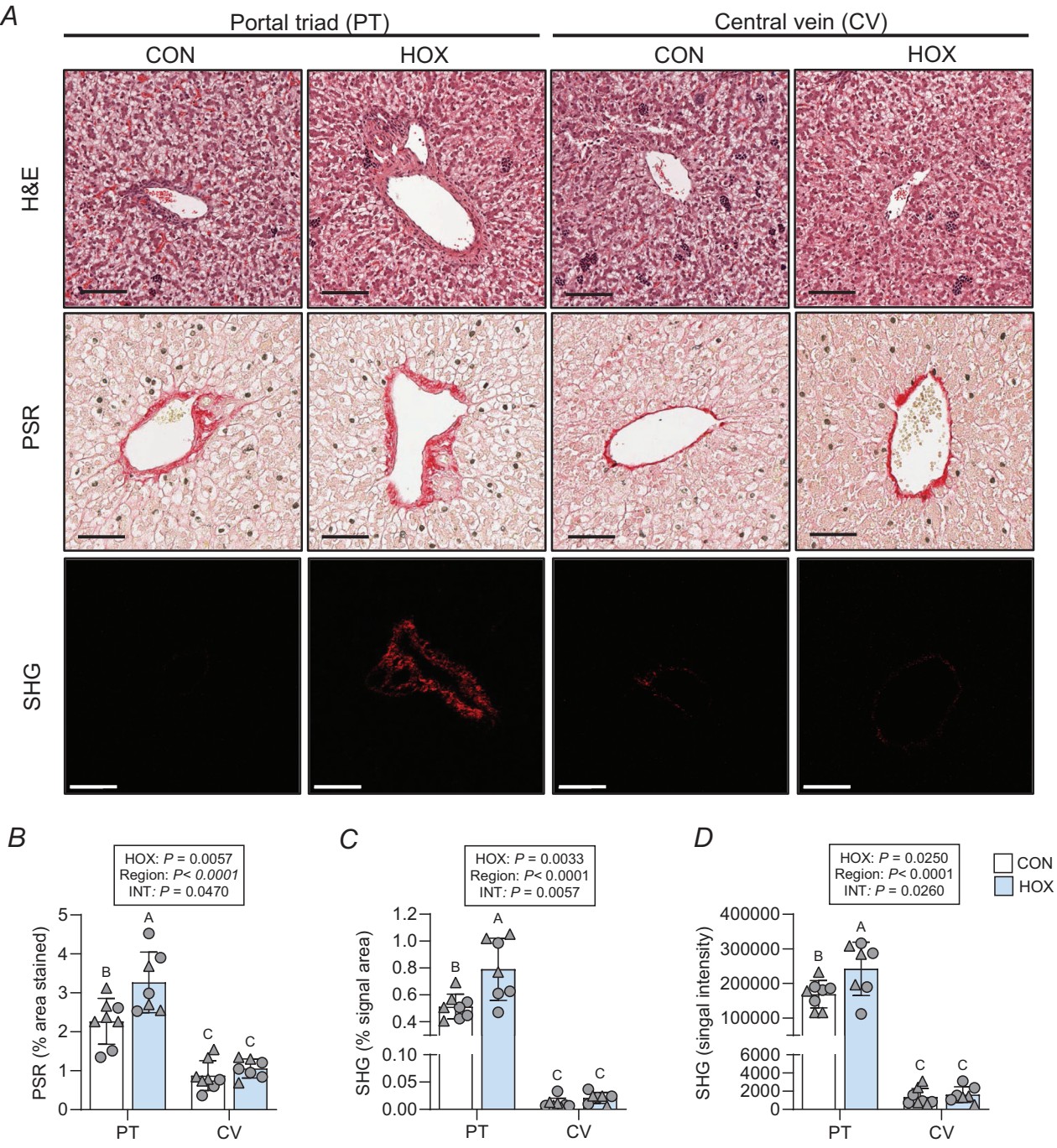

**Figure 1. Hypoxia increases fetal hepatic collagen deposition**

Histological analysis of portal triad (PT) and central vein (CV) regions in formalin-fixed and paraffin-embedded liver sections from normal (CON, $n = 9$) and hypoxic (HOX, $n = 7$) fetuses during late gestation. *A*) representative images of liver sections stained with haematoxylin and eosin (H&E) to assess gross morphology, stained with picrosirius red (PSR) for total collagen, and imaged using second harmonic generation (SHG) to quantify fibrillar collagen. Scale bars = 100 μm. *B*) percent area of PSR staining per region. *C*) percent SHG signal area per region. *D*) SHG signal intensity per region. About 10–15 CV and PT regions were quantified and averaged per animal. Values are the mean ± SD. Results were analysed by ANOVA and *P* values from main effects are shown. Multiple comparisons were performed using Tukey's *post hoc* tests. Different letters indicate statistically significant means. Fetal sex is indicated by triangles for males and circles for females. [Colour figure can be viewed at wileyonlinelibrary.com]

area (Fig. 1*C*) and intensity were greater in PT regions in HOX compared to the CON group (Fig. 1*D*) and there were no differences in CV regions between groups.

## Fetal hepatic metabolites

We next performed semi-targeted metabolomics. Multivariate PLS-DA separated CON and HOX liver samples (Fig. 2*A*). Metabolites with the highest VIP rank scores are shown by heatmap (Fig. 2*B*). HOX livers had increased abundance of amino acids valine and glutamate and decreased abundance of serine, phenylalanine and glycine. Metabolites involved in glucose metabolism included increased D-fructose 1-6-bisphosphate, D-glyceraldehyde 3-phosphate and 2/3-phospho-D-glycerate, and decreased D-glucose six phosphates in HOX livers. There were decreased nucleotides, including GMP, AMP and UMP in HOX liver. HOX livers had increased abundance of biliverdin, 5-oxyproline and 2-hydroxyglutarate, and decreased abundance of gamma-glutamyl-SeMC, and *S*-adenosyl-L-methionine (SAM) in HOX livers, supporting an upregulation of stress-related metabolites and downregulation of antioxidants and metabolites in one-carbon metabolism. There was an increased abundance of sphingolipids in HOX liver, including sphingosine, sphingosine 1-phsophate and ethanolamine phosphate. Notably, HOX livers had increased short-chain (C3–C5) and long-chain (C14–C20) acylcarnitine species compared to CON liver. However, hepatic triglyceride content was not different between groups ($P = 0.1508$) (Fig. 2*C*). Furthermore, lipid droplets were not histologically visible (Fig. 1) and triglyceride levels were relatively low (1.39–2.27 µg mg$^{-1}$ tissue) compared to other models where fetal hepatic steatosis is present (hepatic triglyceride contents of ∼10 µg mg$^{-1}$) (Nash et al., 2021; Wesolowski et al., 2018). A complete list of metabolites is provided in the Supporting information (Table S1).

## Transcriptomics

To determine how hypoxia affected pathways and putative upstream regulators, we performed bulk RNA sequencing and functional analysis. We identified 487 DEGs, with 324 being downregulated and 163 upregulated in HOX compared to CON fetal livers (Fig. 3*A*). Expression of genes related to glutathione synthesis, including glutathione synthase (*GSS*), glutathione *S*-transferase omega 1 (*GSTO1*), microsomal glutathione S-transferase (*MGST1*) and isocitrate dehydrogenase 1 (*IDH1*), were decreased, along with nuclear protein 1 (*NUPR1*), a stress regulator. Glycine, serine and glutamine are important amino acids in glutathione synthesis and

one-carbon metabolism. In HOX livers, there was increased glutamine transporter *SLC38A2* and decreased glycine transporter *SLC6A9* expression. There was also decreased expression of phosphoserine aminotransferase 1 (*PSAT1*) and phosphoserine phosphatase (*PSPH*), comprising two genes in serine biosynthesis. Genes related to cell redox homeostasis were regulated including increased expression of aldehyde oxidase 1 (*AOX1*), an enzyme that catalyses the formation of superoxide, and decreased expression of antioxidant/reactive oxygen species (ROS) detoxification genes including thioredoxin 2 (*TXN2*), glutaredoxin (*GLRX*), phospholipase A2 group XIIB (*PLA2G12B*) and phospholipase D1 (*PLD1*). The expression of erythropoietin receptor (*EPOR*), a known HIF target, and sulfotransferase family 2A member 1 (*SULT2A1*), a regulator of bile acid metabolism, also increased. Other DEGs are shown in the Supporting information (Table S2).

We performed functional analysis of the DEGs and identified 44 enriched canonical pathways (Fig. 3*B*). Notable pathways included those associated with cell death and homeostasis, including a predicted upregulation of granzyme A signalling ($P < 0.001$) and chaperone-mediated autophagy ($P = 0.0191$). Predicted pathways involved in stress and mitochondrial metabolism included glucocorticoid signalling ($P = 0.0398$), upregulation of mitochondrial dysfunction ($P < 0.001$) and downregulation of oxidative phosphorylation ($P < 0.001$) and detoxification of ROS ($P = 0.0166$). Several predicted pathways were involved in energy and nucleotide metabolism, including nucleotide catabolism ($P = 0.00603$) and sirtuin signalling ($P = 0.00776$). A complete list of predicted pathways is listed in the Supporting information (Table S3). We also performed a predicted upstream regulator analysis to determine putative factors that may mediate the effects of hypoxia on gene regulation (Fig. 3*C*). Notable predicted activated regulators involved in mitochondrial function, cell signalling and epigenetic modification were identified and included KDM5A, EPO, KRAS, MAP4K4, SIRT3, CLPP, KDM1A and JUN. Predicted inhibited regulators included factors directly involved in the integrated stress response such as ATF4, NFE2L2 and EIF2AK2, as well as those indirectly related including HNF4, XPB1, SREBF1 and LONP1.

We performed a joint pathway analysis using the regulated metabolites and DEGs (Fig. 3*D*). Predicted impacted pathways of interest included protein export ($P < 0.001$), glycine, serine and threonine metabolism ($P < 0.001$), protein processing in endoplasmic reticulum ($P = 0.00232$), oxidative phosphorylation ($P = 0.00988$), glutathione metabolism ($P = 0.0111$), purine metabolism ($P = 0.0147$), central carbon metabolism ($P = 0.0191$), peroxisome ($P = 0.0248$) and pyrimidine metabolism

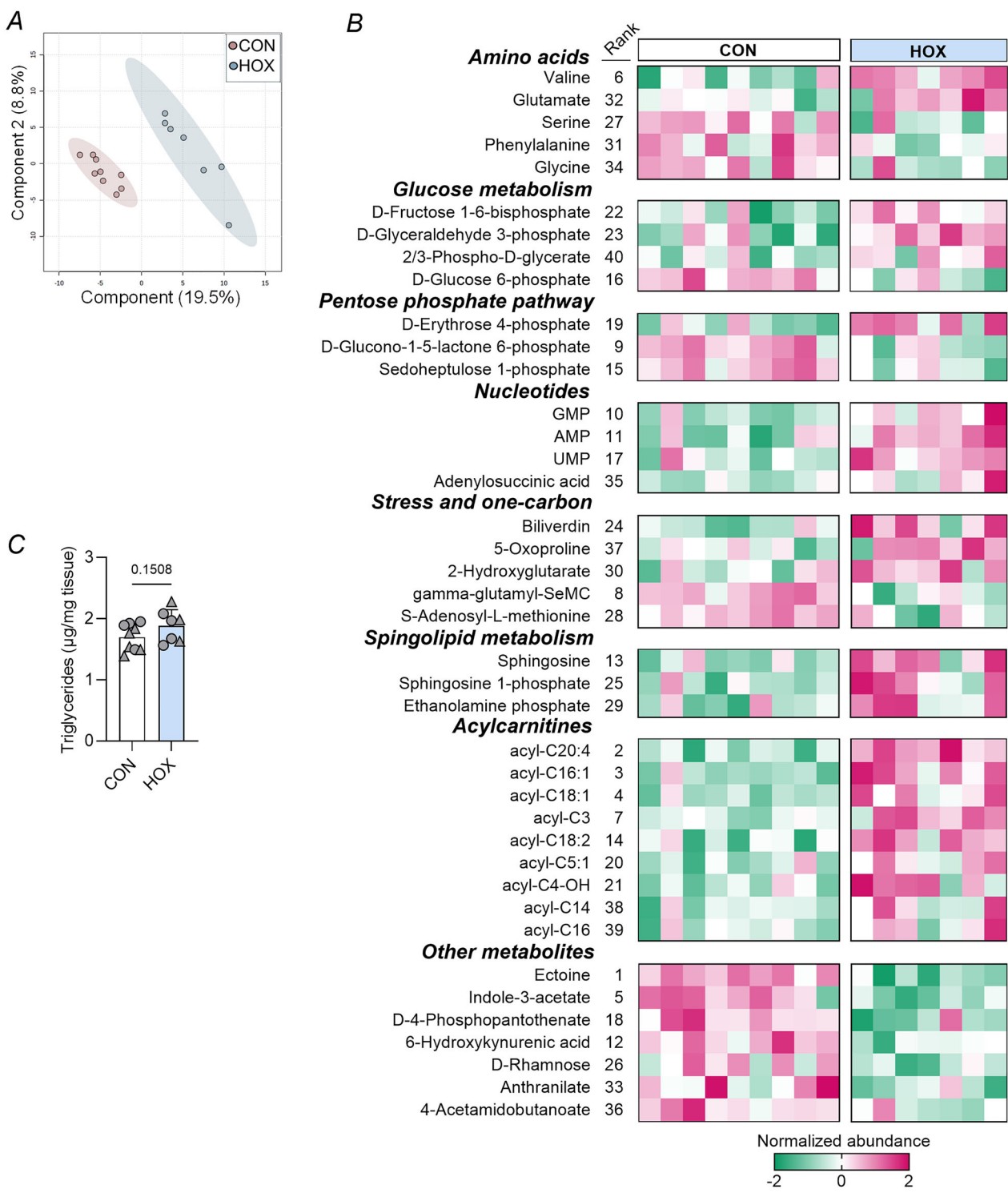

**Figure 2. Hypoxia shifts fetal hepatic metabolites**
Metabolomics analysis was performed to distinguish hepatic metabolite profiles from normal (CON, $n = 9$) and hypoxic (HOX, $n = 7$) fetuses. *A*) groups separated by partial least squares discrimination analysis (PLS-DA) of liver metabolites. *B*) heat map depicting metabolites with the 40 highest ranking variable importance in projection (VIP) scores between groups. Each column represents an animal, and colours indicate normalized abundance for each metabolite. *C*) Triglyceride content in fetal livers. Values are means ± SD. P values calculated by Student's t test. Fetal sex is indicated by triangles for males and circles for females. [Colour figure can be viewed at wileyonlinelibrary.com]

($P = 0.0350$). Other pathways are shown in the Supporting information (Tables S4 and S5).

## Glucose metabolism

Given our metabolic and transcriptomic results supporting effects on central carbon and glucose metabolism, along with prior results in models of placental insufficiency or shorter 10 day hypoxia

exposures (Brown et al., 2024; Jones et al., 2019, 2022), we next measured targets regulating hepatic glucose production, pyruvate oxidation and lactate production (Fig. 4*A*). Relative to CON, HOX livers had higher protein abundance of gluconeogenic enzymes (Fig. 4*B*) PEPCK-C (cytosolic phosphoenol-pyruvate carboxykinase, 1.48-fold, $P = 0.0329$) and G6PC (catalytic subunit of glucose-6-phosphatase, 1.26-fold, $P = 0.0668$). Pyruvate dehydrogenase controls the

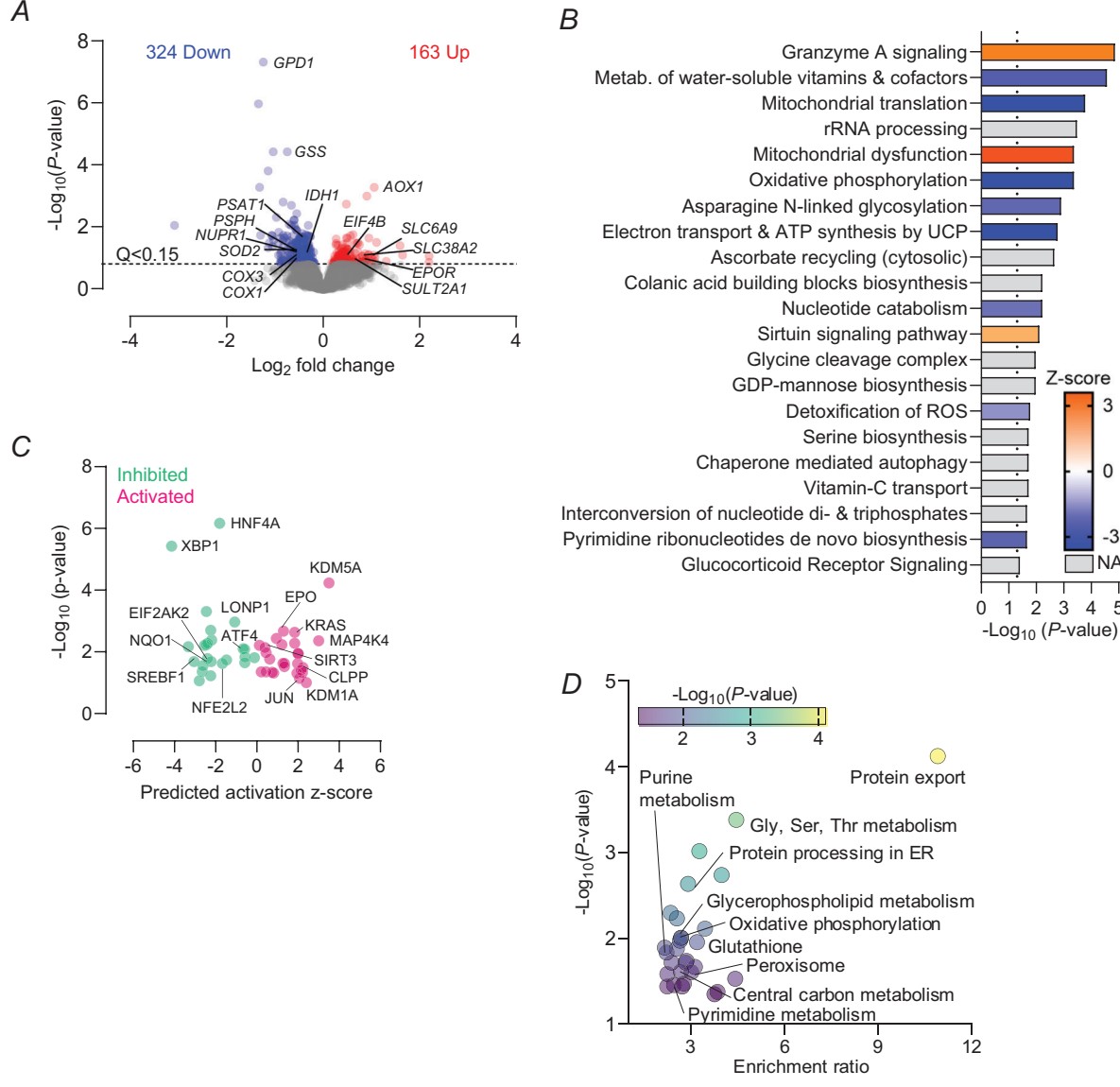

**Figure 3. Hypoxia regulates fetal hepatic gene expression**
Bulk RNA-sequencing was performed on liver from normal (CON, *n* = 9) and hypoxic (HOX, *n* = 7) fetuses. *A*) volcano plot indicating upregulated (red) and downregulated (blue) differentially expressed genes (DEGs) with false discovery rate (*Q* value) < 0.15. *B*) relevant predicted canonical pathways enriched in DEGs. Positive *Z*-score indicates predicted pathway upregulation and negative *Z*-scores indicate predicted downregulation. Grey bars indicate a *Z*-score was not assigned to the predicted pathway. *C*) predicted upstream regulators. Positive *Z*-scores indicates predicted upregulation, and negative *Z*-scores indicates predicted inhibition. *D*) predicted metabolic pathways enriched in DEGs and metabolites with the 40 highest ranking variable importance in projection (VIP) scores between groups. [Colour figure can be viewed at wileyonlinelibrary.com]

oxidation of pyruvate to acetyl-CoA. HOX livers had increased abundance of phosphorylated PDH (ph-PDH$^{Ser5}$, 1.167-fold, $P = 0.0606$) (Fig. 4$C$) at a site that decreases PDH activity and increased total PDH abundance (1.347-fold, $P = 0.0193$). However, the ratio of ph-PDH$^{Ser5}$ to total PDH protein was decreased by 14% ($P = 0.0606$). Congruent with the relative decrease in inhibitory PDH phosphorylation, PDH enzyme activity was increased (1.87-fold, $P < 0.001$) (Fig. 4$D$). HOX livers also had higher LDHA protein abundance (2.24-fold, $P = 0.0186$) (Fig. 4$E$) and LDH activity (1.14-fold, $P = 0.0598$) (Fig. 4$F$).

### Nutrient and stress sensor signalling

Compared to CON, HOX liver had increased activating phosphorylation of AMP-activated protein kinase (AMPK, ph-AMPK$^{T172}$; 1.41-fold, $P = 0.0291$) (Fig. 5$B$), consistent with higher abundance or AMP. Total AMPK (1.21-fold, $P = 0.0949$) and the ratio of ph-AMPK to total AMPK ($P = 0.365$) were not different. The phosphorylation of mTOR (ph-mTOR$^{S2448}$; 1.62-fold,

$P = 0.0433$) (Fig. 5$C$) and the ratio of ph-mTOR to total mTOR (1.38-fold, $P = 0.0384$) was also increased in HOX compared to CON livers, with no difference in total mTOR protein abundance (1.26-fold, $P = 0.0934$). Consistent with increased mTOR phosphorylation and activation, the phosphorylation of eukaryotic translation initiation factor 4E binding protein 1 (ph-4EBP1$^{Thr39/49}$), a downstream target of mTOR, was increased when expressed as a ratio of phosphorylated to total 4EBP1 protein abundance (1.46-fold, $P = 0.0354$) (Fig. 5$D$). However, there was no increase in the phosphorylation of S6, another downstream target of mTOR (Fig. 5$E$).

### Mitochondrial oxidation capacity and oxidative stress

We next sought to evaluate the predicted effects from omics analyses on oxidative metabolism, mitochondrial dysfunction and oxidative stress. There was no difference in the abundances of proteins associated with oxidative phosphorylation complexes I ($P = 0.411$), II ($P = 0.400$), III ($P = 0.504$), IV ($P = 0.301$) or V ($P = 0.567$) (Fig. 6$A$ and $B$). There was also no difference in TBARS, a measure

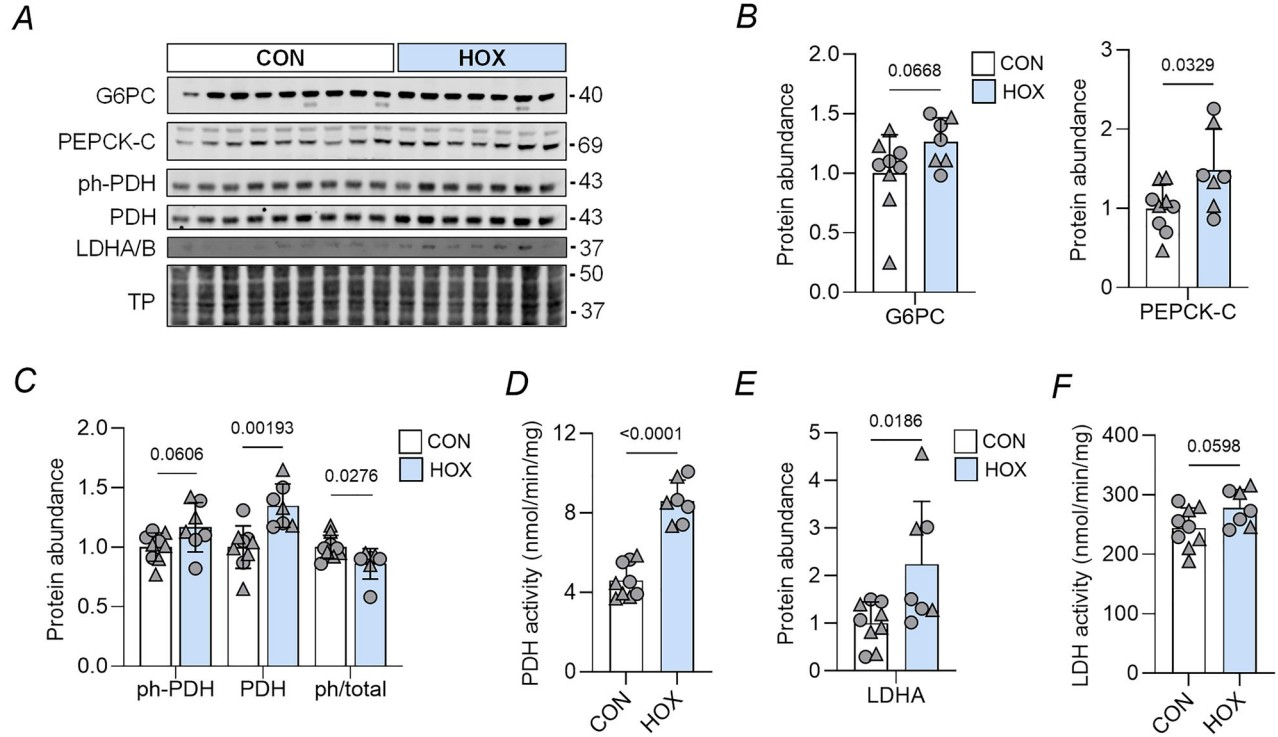

**Figure 4. Hypoxia regulates carbohydrate metabolic pathways in the fetal liver**
Protein abundance and enzyme activity was measured in liver from normal (CON, $n = 9$) and hypoxic (HOX, $n = 7$) fetuses. *A*) western blot images for proteins involved in carbohydrate metabolism and total protein stain (TP). Molecular weights for protein bands are indicated. *B*) protein abundance of glucose-6-phosphatase catalytic subunit 1 (G6PC) and cytosolic phosphoenolpyruvatecarboxykinase (PEPCK-C). *C*) protein abundance of total pyruvate dehydrogenase (PDH), phosphorylated PDH$^{Ser\ 293}$ (ph-PDH) and the ratio of ph-PDH to total PDH (ph/total). *D*) PDH enzyme activity. *E*) protein abundance of lactate dehydrogenase A (LDHA). *F*) LDHA enzyme activity. Values are the mean ± SD. *P* values calculated by Student's *t* test. Fetal sex is indicated by triangles for males and circles for females. [Colour figure can be viewed at wileyonlinelibrary.com]

of lipid peroxidation and oxidative stress ($P = 0.107$) (Fig. 6*C*), the ratio of glutathione (GSH) to oxidized glutathione (glutathione disulfide, GSSG) ($P = 0.598$) or urate, a marker of oxidative stress in the fetus (Kane et al., 2014) in liver tissue as measured in our metabolomics data (Fig. 6*D*). We also calculated the glutamate-serine-glycine (G-S-G) index, an indicator associated with metabolic liver disease and fibrosis (Gaggini et al., 2018), which was 2.23-fold greater in HOX liver than CON ($P = 0.0033$) (Fig. 6*E*).

To assess antioxidant response, we measured NRF2, a ROS sensor and activator of the antioxidant response element, and two downstream targets (Fig. 6*F* and *G*). There was no difference in NRF2 protein abundance ($P = 0.564$) or its downstream targets NQO1 ($P = 0.501$), an enzyme that prevents ROS by reducing quinones, or the mitochondrial antioxidant SOD2 ($P = 0.375$), despite decreased *SOD2* gene expression in HOX livers.

### Relationships between fetal hepatic collagen deposition and stress metabolites

We performed a correlation analysis to assess the putative causal or consequential roles of fetal hypoxia and stress-related liver metabolites on increased hepatic collagen deposition (Fig. 7*A*). We used PSR staining and SHG signal area, which were positively correlated ($r = 0.057$, $P = 0.0344$), to represent hepatic collagen deposition. PSR was positively correlated with blood haemoglobin and liver tissue urate, AMP, sphingosine and biliverdin levels. SHG was positively correlated with the liver tissue G-S-G index and 5-oxyproline levels. Blood haemoglobin, as a readout of fetal hypoxia, was associated with increased AMP and biliverdin and decreased GSH:GSSG in liver tissue. The full correlation matrix is depicted in the Appendix (Fig. A1).

### Discussion

Hypoxic pregnancy increased fetal hepatic collagen deposition, indicating liver injury, in association with a unique profile of liver stress metabolites and adaptations in central carbon metabolism. The metabolomics and transcriptomics approaches predicted disruptions in mitochondrial dysfunction and decreased oxidative phosphorylation. In support, we found potentiation of the gluconeogenic pathway and increased lactate production, pyruvate oxidation and AMPK activation. By contrast to the predicted effects, hypoxic livers maintained mitochondrial oxidation and antioxidant capacity. Inter-

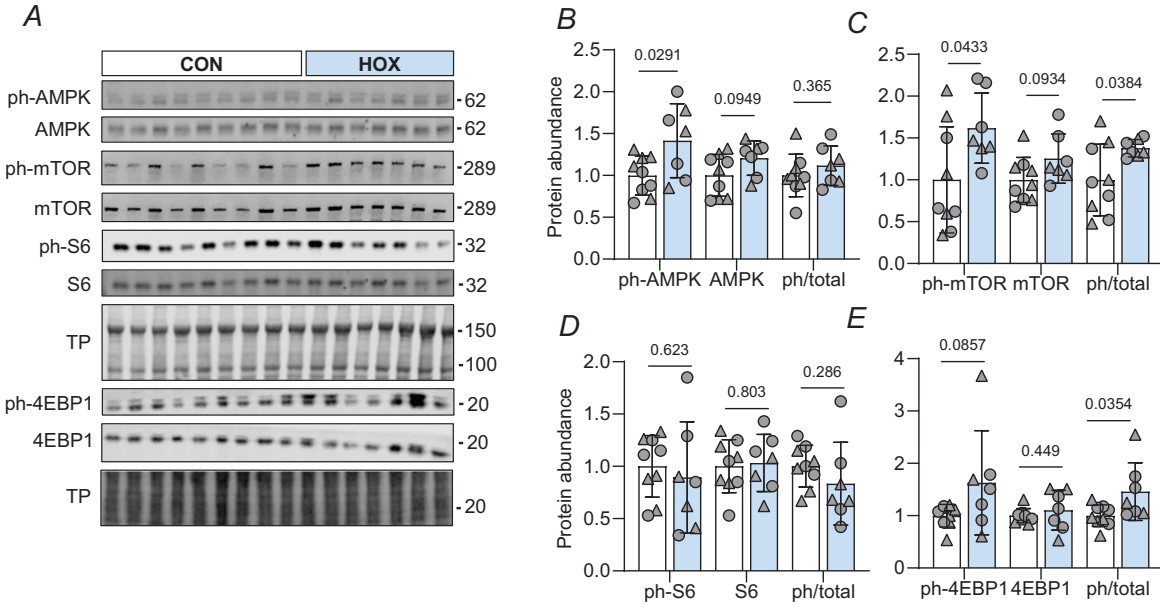

**Figure 5. Effect of hypoxia on energy and nutrient stress signalling in fetal liver**
Protein abundance was measured by western blotting in liver from normal (CON, *n* = 9) and hypoxic (HOX, *n* = 7) fetuses. *A*) western blot images of nutrient and stress sensing proteins and total protein stain (TP) with molecular weight of protein bands indicated. *B*) protein abundance of total APMK, phosphorylated AMPK$^{Thr172}$, and the ratio of phosphorylated to total AMPK (ph/total). *C*) protein abundance of total mammalian target of rapamycin (mTOR), phosphorylated mTOR$^{Ser2448}$, and the ratio of phosphorylated to total mTOR (ph/total). *D*) protein abundance of total ribosomal protein S6 (S6), phosphorylated S6$^{Ser\ 235/236}$ and the ratio of phosphorylated to total S6 (ph/total). *E*) protein abundance of total eukaryotic translation initiation factor 4E binding protein 1 (4EBP1), phosphorylated 4EBP1$^{Thr39/49}$ and the ratio of total to phosphorylated 4EBP1. [Colour figure can be viewed at wileyonlinelibrary.com]

estingly, acylcarnitines were increased, yet hepatic triglyceride content was similar. Furthermore, there was little indication of activation of established oxidative damage stress markers, such as lipid peroxidation or oxidized glutathione. However, we uncovered a unique profile of liver stress-related metabolites in association with periportal collagen deposition (Fig. 7*B*). Together, these adaptations may support a metabolic strategy that enables the hypoxic fetus to withstand hypoxia *in utero* and protect against overt oxidative damage yet still initiate

liver collagen deposition without increased triglyceride accumulation.

Our results extend prior studies supporting that hypoxia *in utero* induces liver injury in exposed offspring. Fetal guinea-pigs exposed to hypoxia had increased hepatic lipid peroxidation and DNA fragmentation, signs of injury, which was ameliorated by treatment with *N*-acetylcysteine, a glutathione precursor (Hashimoto et al., 2012). In a guinea-pig model of placental insufficiency with growth restriction, postnatal offspring

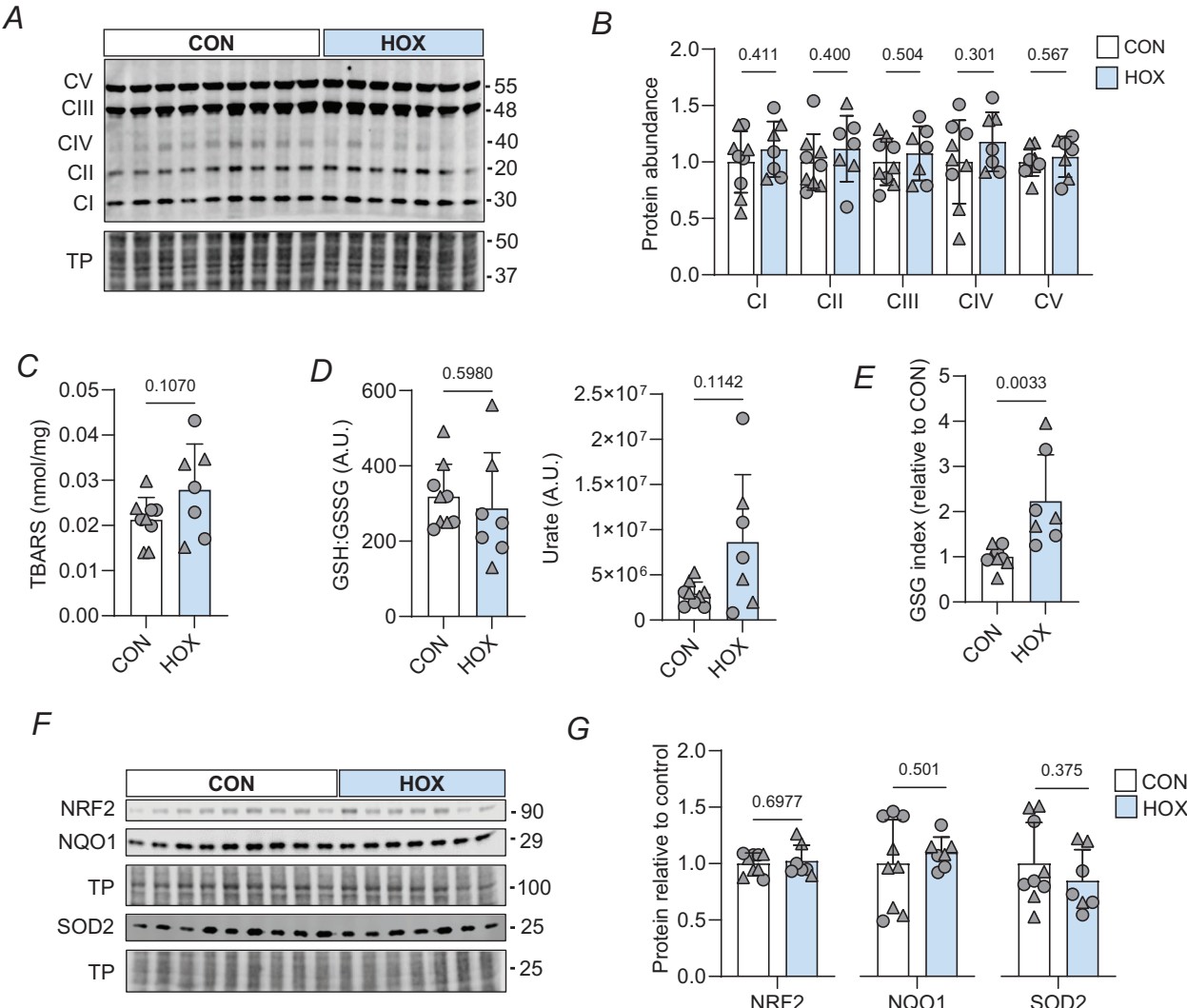

**Figure 6. Effect of hypoxia on mitochondria and oxidative stress pathways in the fetal liver**
Protein abundance, lipid peroxidation, and metabolites were measured in liver obtained from normal (CON, *n* = 9) and hypoxic (HOX, *n* = 7) fetuses. *A*) western blot image for oxidative phosphorylation (OXPHOS) complexes I–V and total protein stain (TP) with molecular weights of protein bands indicated. *B*) protein abundance of OXPHOS proteins. *C*) hepatic thiobarbituric acid-reactive substances (TBARS) content. *D*) ratio of hepatic glutathione to glutathione disulfide (GSH:GSSG) and urate content. *E*) hepatic glutamate-serine-glycine index. *F*) western blot images of antioxidant response proteins and TP with molecular weights of protein bands indicated. *G*) protein abundance of nuclear factor-erythroid 2-related factor 2 (NRF2), NADPH quinone dehydrogenase 1 (NQO1) and superoxide dismutase 2 (SOD2). Values are the mean ± SD. *P* values calculated by Student's *t* test. Fetal sex is indicated by triangles for males and circles for females. [Colour figure can be viewed at wileyonlinelibrary.com]

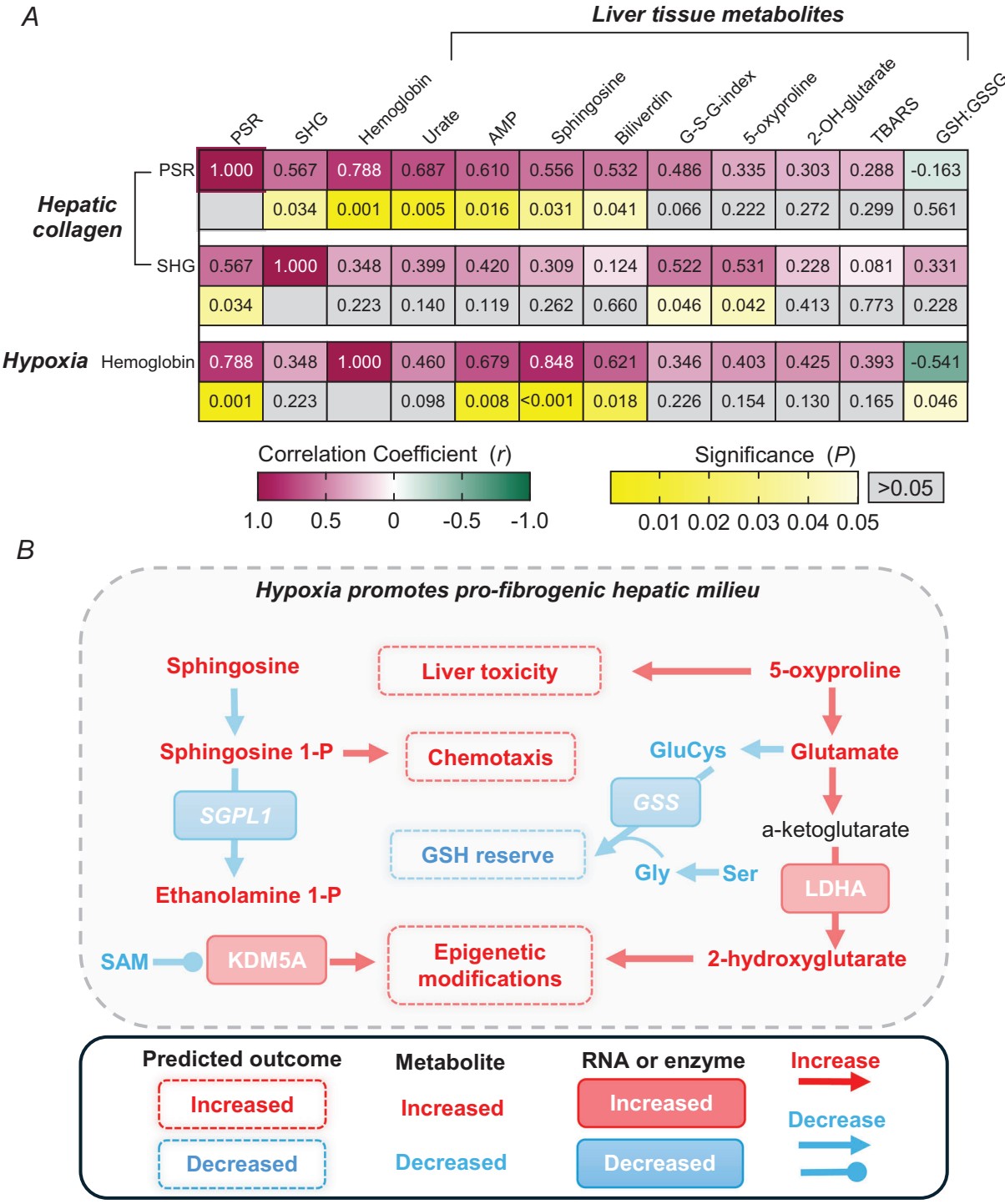

**Figure 7. Relationship between hepatic collagen, hypoxia, and stress markers in late gestation fetal liver**

*A*) correlations between hepatic collagen measured by percent staining area of picrosirius red (PSR) and second harmonic generation (SHG) signal area in periportal regions, plasma haemoglobin, and hepatic metabolites indicative of stress including glutamate-serine-glycine (G-S-G-index), lipid peroxidation (TBARS) and glutathione to glutathione disulfide (GSH:GSSG). Values indicate Pearson correlation coefficients ($r$) and $P$ values. Measurements were used from normal (CON, $n = 9$) and hypoxic (HOX, $n = 7$) fetal livers. *B*) diagram showing relationship between regulated RNA, enzymes and metabolites along with predicted outcomes in hypoxic fetal liver. GSH, glutathione; Gly, glycine; Ser, serine; GluCys, glutamly-cysteine; *GSS*, glutathione synthase; LDHA, lactate dehydrogenase A, *SGPL1*, sphingosine-1-phosphate lyase 1; KDM5A, lysine demethylase 5A; SAM, *S*-adenosyl-L-methionine. [Colour figure can be viewed at wileyonlinelibrary.com]

exposed to hypoxia *in utero* had increased hepatic collagen deposition and more severe hepatic steatosis after post-natal western-style diet exposure (Sarr et al., 2016). In non-human primates, chronic maternal western-style diet exposure increased hepatic collagen deposition in the third-trimester fetus with persistence in the offspring at 3 years of age (Nash et al., 2021, 2023). Interestingly, these fetuses also had lower oxygenation associated with increased hepatic collagen deposition, and an intervention with the antioxidant resveratrol lowered collagen deposition (Nash et al., 2021). Indeed, children born with FGR are at increased risk of hepatic steatosis and fibrosis, hallmarks of metabolic-associated steatotic liver disease (MASLD) (Alisi et al., 2011; Nobili et al., 2016, 2007). Further, among children with biopsy confirmed MASLD, those with lower birth weights had higher fibrosis scores (Alisi et al., 2011). Our data show increased collagen, an early sign of liver injury, that presents in late gestation in the hypoxic fetus without steatosis. Combined, these data suggest that *in utero* hypoxia promotes a profibrogenic environment in the fetal liver with long-term consequences for liver fibrotic disease.

Interestingly, the localization of collagen within periportal *vs.* pericentral regions of the fetal liver is a distinguishing feature of paediatric compared to adult MASLD (Liu et al., 2017; Nobili et al., 2016; Schwimmer et al., 2005). In the fetus, the periportal region receives the first pass of umbilical venous blood coming from the placenta. Therefore, lower oxygen supply and associated factors entering the fetal liver during pregnancies with hypoxia may initiate periportal fibrosis. Further work is warranted to understand the interplay between oxygen supply, systemic factors and whether localized production of the stress metabolites (Fig. 7*B*) contributes to the periportal distribution of collagen deposition in the hypoxic fetal liver. Moreover, we found that increased collagen deposition occurred without steatosis. Thus, mechanisms initiating fibrosis in the fetal liver may be mediated by hypoxia *in utero* and differ from those involved in the classical progression from steatosis to MASH in adult MASLD (Liu et al., 2017).

Additionally, hypoxia impacted amino acids, glucose, pentose phosphate and one-carbon metabolism. In the normally growing fetus, the liver utilizes lactate and amino acids for oxidative metabolism and releases glutamate, serine and pyruvate (Brown et al., 2024). In fetal sheep, serine requirements are met by endogenous production from glycine in the liver (Cetin et al., 1991; Thureen et al., 1995). HOX fetuses had decreased hepatic glycine and serine and increased glutamate content. Thus, the decrease in serine is probably a result of decreased glycine precursor and decreased expression of serine synthesis genes *PSPH* and *PSAT1*. Increased glutamate suggests increased retention and utilization of carbon atoms for anabolic and biosynthetic pathways,

potentially at the expense of oxidation, in the HOX fetal liver. Furthermore, in HOX fetuses, there was increased protein levels of gluconeogenic enzymes G6PC and PEPCK-C and increased abundance of glucose intermediates D-fructose 1-6-bisphosphate, D-glyceraldehyde 3-phosphate and 2/3-phospho-D-glycerate, yet lower glucose 6-phosphate and no change in hepatic glucose. Thus, hypoxia may potentiate a truncated gluconeogenic pathway that permits intermediates to be generated and used for other biosynthetic pathways. Moreover, in the normal fetus, hepatic glucose production is not active, although it can be activated in response to hypoglycaemia (Houin et al., 2015) and chronic placental insufficiency (Thorn et al., 2013). Because HOX fetuses were not hypoglycaemic, the need to activate hepatic glucose production was not present. There were also shifts in metabolites in the pentose phosphate pathway and one carbon metabolism such as increased D-erythrose 4-phosphate, as well as decreased D-glucono-1-5-lactone 6-phosphate and sedoheptulose 1-phosphate in HOX livers. We also found increased acylcarnitines, however, without a decrease in abundance of oxidative phosphorylation proteins, and this may reflect decreased oxygen supply to fuel oxidation, rather than a defect in mitochondrial capacity. Moreover, lipids are minimal oxidative fuel in the fetal liver and *de novo* lipogenesis rates are low (Bell & Freetly, 2005; Girard et al., 1979). This may explain the absence of triglyceride accumulation.

In the present study, hypoxic pregnancy increased AMP content and AMPK phosphorylation in the fetal liver. AMPK mediates the cellular response to energetic stress and promotes mitochondrial health (Herzig & Shaw, 2018). Activated AMPK may help hypoxic fetuses maintain hepatic mitochondrial capacity. Additionally, AMPK activation maintains PDH activity and pyruvate oxidation via phosphorylation at unique sites (S314) that overrides inhibitory phosphorylation by pyruvate dehydrogenase kinases (PDK) at S293 (Cai et al., 2020). HOX fetuses had increased hepatic PDH activity, congruent with the increased activation of AMPK. Furthermore, our transcriptomics analysis revealed no decrease in *PDK*1, *PDK*2 or *PDK*4 expression in HOX liver, supporting that increased PDH activity could be attributed to AMPK activation rather than reduced PDK inhibition. AMPK also inhibits mTORC1 signalling (Saxton & Sabatini, 2017). Although mTOR phosphorylation at site Ser2448 was increased in the HOX liver, only phosphorylation of one of its downstream targets, 4EBP1, was increased. There was no increase in phosphorylation of S6. This suggests a possible impairment in nutrient sensing or a disconnect between AMPK and mTOR signalling (Saxton & Sabatini, 2017).

Fetal hypoxia increased collagen deposition without classic oxidative stress markers and steatosis. Oxidative stress is a driver of hepatic fibrosis and activation of

antioxidant responses protects against tissue damage (Koyama & Brenner, 2017; Trivedi et al., 2021). Interestingly, we found no increase in TBARS, GSH or urate, which are established oxidative stress markers (Parola & Robino, 2001), nor evidence supporting decreased mitochondrial oxidation or antioxidant capacity. Rather, increased collagen deposition in the HOX liver occurred in parallel with a unique profile of hypoxia- and stress-related metabolites, including biliverdin, 5-oxoproline, 2-hydroxyglutarate, gamma-glutamly-SeMC and *S*-adenosyl-ʟ-methionine. Several elevated metabolites in the HOX liver are linked to inflammation and liver injury (Fig. 7*B*). For example, an elevated GSG index is a biomarker for MASLD in adults (Gaggini et al., 2018) and overweight adolescents (Leonetti et al., 2020). HOX livers also had increased sphingosine and sphingosine-1-phosphate, which have been implicated in liver disease and promote chemotaxis of collagen-producing hepatic stellate cells (Ishay et al., 2020; Lan et al., 2018; Lee et al., 2023). Additionally, 2-hydroxyglutarate is elevated in newborns with biliary atresia (Tian et al., 2022) and some cancers (Qing et al., 2021). Furthermore, 2-hydroxyglutarate is induced under hypoxic conditions by LDHA (Intlekofer et al., 2015; Tian et al., 2022) and provides redox buffering reducing equivalents in cells (Oldham et al., 2015). This suggests an intriguing hypothetical pathway involving increased LDHA activity and 2-hydroxyglutarate in the development of fibrosis. The methyl donor SAM was decreased in HOX. SAM inhibits the histone methylase KDM5A, a predicted upstream regulator in our transcriptomics data (Fig. 7*B*). Also, 2-hydroxyglutarate regulates epigenetic modifications by inhibiting histone demethylases (Ye et al., 2018). Combined, the data identify a distinct profile of stress metabolites in HOX livers that may contribute to a fibrogenic environment and underlie potential epigenetic programming.

Endogenous antioxidant capacity protects against increased ROS production during late gestation when fetal hepatic oxygen consumption increases (Thibeault, 2000). Hypoxic guinea pig fetuses had a two-fold increase in liver glutathione content after 14 day hypoxic exposure (Oh et al., 2008). In the present study, HOX livers had decreased expression of *GSS* and glutathione precursors gamma-glutamyl cysteine, glycine and serine and increased 5-oxyproline, a product of glutathione breakdown that can be toxic at elevated levels (Spielberg et al., 1977; Verma et al., 2012). This suggests the consumption of glutathione reserves (Liu et al., 2014) (Fig. 7*B*). Antioxidant depletion during the fetal period may increase the risk of oxidative damage postnatally (Thompson & Al-Hasan, 2012). Placental insufficiency-induced FGR fetal sheep have reduced liver glutathione (Jones

et al., 2019), increased liver TBARS associated with increased AMPK activation and higher AMP:ATP and lower NADH:NAD+ (Brown et al., 2024), supporting energetic and oxidative stress. Compared to adults, the fetus has lower antioxidant capacity (Asikainen et al., 1998; Elbarbry & Alcorn, 2009). During the final 10% of gestation, antioxidant enzymes such as SOD, catalase and glutathione peroxidase increase in the fetal lung in preparation for ambient oxygen levels (Frank & Sosenko, 1987). Although no changes were observed in oxidative stress markers or antioxidant genes *SOD1* or *SOD2* in HOX fetal lungs, there was an increase in antioxidant *CAT* and decreased pro-oxidant *NOX4* in the HOX fetal lung (McGillick et al., 2017). Furthermore, the placenta of the HOX sheep in this study showed evidence of oxidative stress and activation of the unfolded protein response (Tong et al., 2022).

A limitation of the present study is that all measures are obtained at a single gestational time point. The duration of hypoxic insults and the stage of gestation at which they occur warrants further investigation across tissues to help understand how stress and metabolic effects develop and adapt over time. Furthermore, our model produced fetal hypoxia via induction of maternal hypoxia, which is highly relevant for understanding humans living at altitude. However, maternal hypoxia may produce additional effects that are not present in other pregnancy conditions, such as placental insufficiency, where the mother remains normoxic and asymptomatic. Indeed, maternal hypoxia initiated at 70% gestation, as in the present study, may limit placental function and nutrient transport to the fetus, contributing to growth restriction. Additional studies are needed to investigate this. We also acknowledge that we used an FDR-adjusted *P* value, $Q < 0.15$, in our transcriptomics analysis to maximize the number of DEGs identified. By performing functional annotation analyses of these DEGs, this represents a robust approach to identify predicted regulated pathways, alone and with our metabolomics data.

In summary, offspring born with FGR have an increased risk of developing liver metabolic disease, including MASLD, fibrosis, biliary atresia, cholestasis and insulin resistance across their lifespan (Eriksson, 2019; Feldman & Sokol, 2019; Giussani, 2021; McMillen & Robinson, 2005; Wesolowski et al., 2017). The data in our study show that hypoxic pregnancy increased fetal hepatic collagen deposition, an indicator of hepatic injury, and triggered a unique profile of liver stress metabolites and adaptations in central carbon metabolism. These data provide new insight into how chronic fetal hypoxia during FGR may trigger a fetal origin of liver metabolic diseases in offspring by initiating fibrosis and disrupting metabolic pathways.

# Appendix

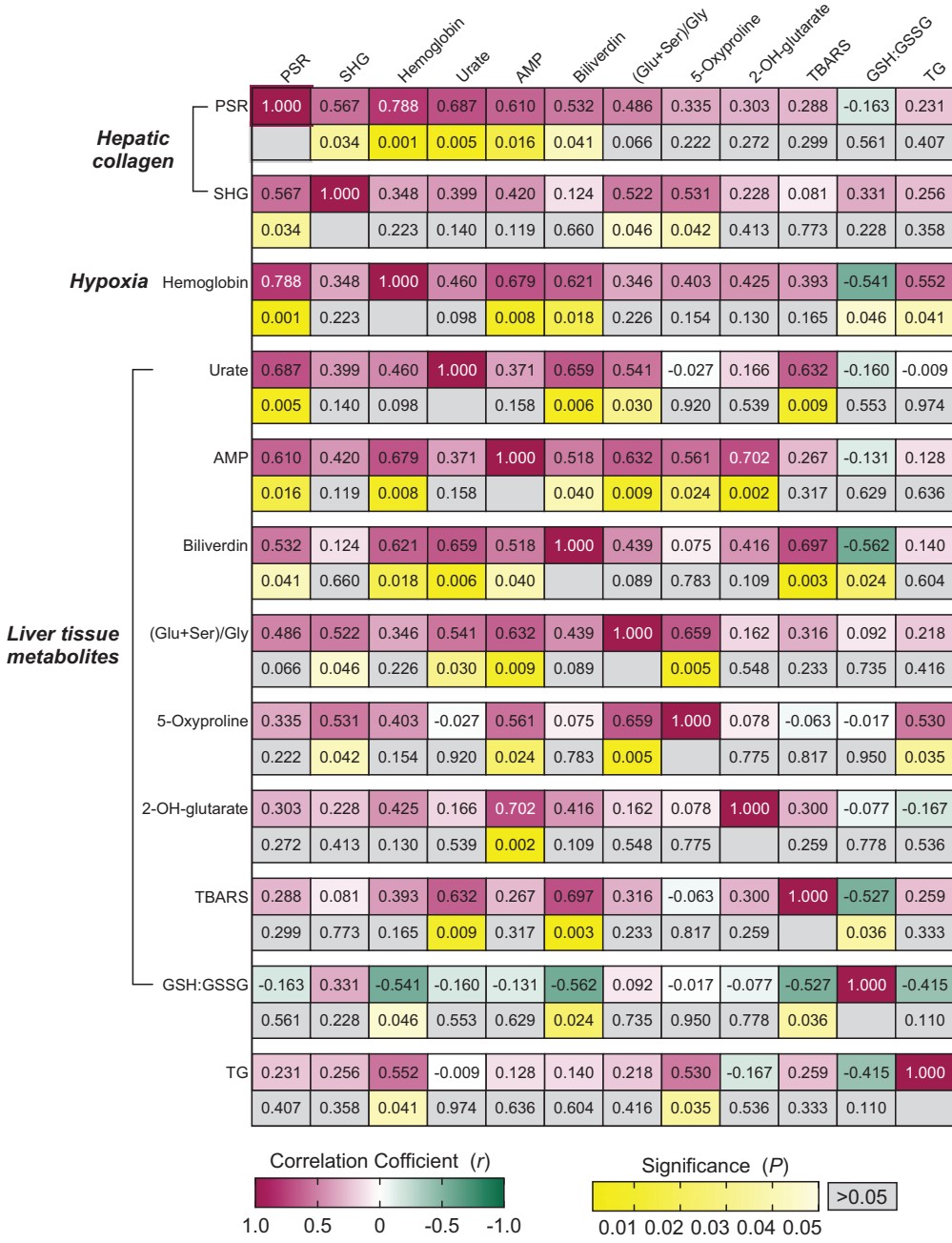

**Figure A1. Relationship between hepatic collagen, hypoxia, and stress markers in late gestation fetal liver**
Correlations between hepatic collagen measured by percent staining area of picrosirius red (PSR) and second harmonic generation (SHG) signal area in periportal regions, plasma haemoglobin and hepatic metabolites indicative of stress including glutamate-serine-glycine (G-S-G-index), lipid peroxidation (TBARS), and glutathione to glutathione disulfide (GSH:GSSG). Values indicate Pearson correlation coefficients (*r*) and *P* values. Measurements were used from normal (CON, *n* = 9) and hypoxic (HOX, *n* = 7) fetal livers. [Colour figure can be viewed at wileyonlinelibrary.com]

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

## Additional information

### Data availability statement

RNA-sequencing data are available on the National Centre for Biotechnology Information's Gene Expression Omnibus (GSE282736, https://www.ncbi.nlm.nih.gov/geo/query/acc.cgi?acc=GSE282736). All other data reported in this study are available within the published article and in the supplementary materials.

### Competing interests

The authors declare that they have no competing interests.

### Author contributions

M.M.M. and S.R.W. designed the liver tissue experiments, analysed data and interpreted the data. M.M.M., S.R.W. and D.A.G. wrote the manuscript. D.A.G., W.T., K.J.B. and Y.N. generated the animal model and provided liver tissue samples. M.M.M., D.W. and J.O. performed assays and analysed data. E.D. assisted with histological analysis. All authors provided data interpretations and reviewed the manuscript.

### Funding

This work was supported by National Institutes of Health grants R01-DK108910 (to SRW), R01-HD134949 (to SRW), F32-DK139657 (to MMM) and T32-HD007186 (to MMM), and the National Institutes of Health P30-CA06934 funded Genomics Shared Resource (RRID:SCR_021984), Cancer Centre Support Grant (P30CA046934) Mass Spectrometry Metabolomics Shared Resource (RRID:SRC_021988), the Rocky Mountain Neurological Disorders Core grant P30-NS048154, and by Diabetes Research Centre grant P30-DK116073. The work was also funded by the British Heart Foundation (RG/17/8/32924 to DAG) and the Loke Centre for Trophoblast Research at the University of Cambridge, UK (WT and DAG).

### Acknowledgements

We thank Richard Benninger and Radu Moldovan (University of Colorado School of Medicine Advanced Light Microscopy Core) for the use of their equipment; Schuyler Lee and Bifeng Gao for performing RNA-sequencing (University of Colorado School of Medicine Genomics Core); and Julie Reisz Hanes and Angelo D'Alessandro for performing metabolomics assays (University of Colorado School of Medicine Metabolomics Core). We also grateful for the technical help provided by the University Biological Services of the University of Cambridge.

### Keywords

fetal, hypoxemia, metabolism

## Supporting information

Additional supporting information can be found online in the Supporting Information section at the end of the HTML view of the article. Supporting information files available:

**Peer Review History**
**Table S1**
**Table S2**
**Table S3**
**Table S4**
**Table S5**

