## [Peer Review History · The Journal of Physiology]

Hypoxic Pregnancy Promotes Fibrosis and Increases Stress Metabolites in the Ovine Fetal Liver

Molly M McGuckin, Dong Wang, Jasmine Ortiz, Evgenia Dobrinskikh, Wen Tong, Kimberley J Botting-Lawford, Youguo Niu, Dino A Giussani, and Stephanie R Wesolowski

DOI: 10.1113/JP288724

Corresponding author(s): Stephanie Wesolowski (stephanie.wesolowski@cuanschutz.edu)

The following individual(s) involved in review of this submission have agreed to reveal their identity: Brendan M Gabriel (Referee #1)

Review Timeline:

Submission Date:	12-Feb-2025
Editorial Decision:	10-Mar-2025
Revision Received:	17-Mar-2025
Accepted:	03-Apr-2025

Senior Editor: Laura Bennet

Reviewing Editor: Kyle McCommis

Transaction Report:

Dear Dr McGuckin,

Re: JP-RP-2025-288724 **"Hypoxic Pregnancy Promotes Fibrosis and Increases Stress Metabolites in the Fetal Liver"**
by Molly M McGuckin, Dong Wang, Jasmine Ortiz, Evgenia Dobrinskikh, Wen Tong, Kimberley J Botting, Youguo Niu, Dino A Giussani, and Stephanie R Wesolowski

Thank you for submitting your manuscript to The Journal of Physiology. It has been assessed by a Reviewing Editor and by 2 expert referees and we are pleased to tell you that it is acceptable for publication following satisfactory revision.

REVISION CHECKLIST:

We look forward to receiving your revised submission.

Yours sincerely,

Laura Bennet
Senior Editor
The Journal of Physiology

REQUIRED ITEMS

- Author photo and profile. First or joint first authors are asked to provide a short biography (no more than 100 words for one author or 150 words in total for joint first authors) and a portrait photograph. These should be uploaded and clearly labelled together in a Word document with the revised version of the manuscript. See Information for Authors for further details.

- You must start the Methods section with a paragraph headed Ethical approval (https://jp.msubmit.net/cgi-bin/main.plex?form_type=display_requirements#methods).

Research must comply with The Journal's policies regarding animal experiments (<https://physoc.onlinelibrary.wiley.com/hub/animal-experiments>) and adherence to these policies must be stated in the manuscript.

Authors should confirm in their Methods section that their experiments were carried out according to the guidelines laid down by their institution's animal welfare committee, including an ethics approval reference number. The Methods section must contain a statement about access to food, water and housing, details of the anaesthetic regime: anaesthetic used, dose and route of administration, and method of killing the experimental animals.

- Please upload separate high-quality figure files via the submission form.

- You must upload original, uncropped western blot/gel images (including controls) if they are not included in the manuscript. This is to confirm that no inappropriate, unethical or misleading image manipulation has occurred. These should be uploaded as 'Supporting information for review process only'. Please label/highlight the original gels so that we can clearly see which sections/lanes have been used in the manuscript figures. For more information, see: <https://physoc.onlinelibrary.wiley.com/hub/journal-policies#imagmanip>.

- Your paper contains Supporting Information of a type that we no longer publish, including supplementary tables and figures. Any information essential to an understanding of the paper must be included as part of the main manuscript and figures. The only Supporting Information that we publish are video and audio, 3D structures, program codes and large data files. Your revised paper will be returned to you if it does not adhere to our Supporting Information Guidelines.

- Please include an Abstract Figure file, as well as the Figure Legend text within the main article file. The Abstract Figure is a piece of artwork designed to give readers an immediate understanding of the research and should summarise the main conclusions. If possible, the image should be easily 'readable' from left to right or top to bottom. It should show the physiological relevance of the manuscript so readers can assess the importance and content of its findings. Abstract Figures should not merely recapitulate other figures in the manuscript. Please try to keep the diagram as simple as possible and without superfluous information that may distract from the main conclusion(s). Abstract Figures must be provided by authors no later than the revised manuscript stage and should be uploaded as a separate file during online submission labelled as

File Type 'Abstract Figure'. Please also ensure that you include the figure legend in the main article file. All Abstract Figures should be created using BioRender. Authors should use The Journal's premium BioRender account to export high-resolution images. Details on how to use and access the premium account are included as part of this email.

EDITOR COMMENTS

Reviewing Editor:

Thank you for submitting your manuscript titled "Hypoxic Pregnancy Promotes Fibrosis and Increases Stress Metabolites in the Fetal Liver" to The Journal of Physiology. It has been assessed by an academic reviewing editor and two external peer reviewers, all of whom believe this is a very nice set of studies and a manuscript that is likely to have an influence on the field. Both reviewers have several specific comments that should be addressed in order to revise and further improve the manuscript.

Please also see 'Required Items' above.

REFEREE COMMENTS

Referee #1:

McGuckin et al. have demonstrated that a hypoxic for pregnant ewes leads to increased foetal hepatic collagen deposition, indicating early liver injury. They observed disruptions in central carbon metabolism, mitochondrial dysfunction, and enhanced gluconeogenesis, while mitochondrial oxidation and antioxidant capacity remained largely intact. I have detailed the strength of the manuscript below. I also have some comments that should be addressed before I can recommend for publication.

Strengths

Timely and important topic.

Comprehensive analysis of samples using advanced techniques.

Mammalian model with more similarities to human physiology than rodent models.

Robust outcomes from an experimental approach that requires high degree of expertise.

Clearly written and robust experimental methods

Figures are clear and informative.

Discussion references relevant literature.

Conclusion is valid and justified by experimental data.

Major comments

I have a major comment on the chosen model: Although I applaud the use of an ovine model, in most cases of foetal chronic hypoxia in humans, the mother remains asymptomatic because her oxygen levels and systemic circulation are sufficient to maintain her own oxygenation, even if oxygen transfer to the foetus is compromised. Whereas in this current model, the pregnant ewes were exposed to environmental hypoxia. This model would arguably be more relevant for humans living at altitude, rather than foetal chronic hypoxia. This limits the relevance of the model to foetal chronic hypoxia in humans and this limitation should clearly be stated in the discussion.

Given that the ewes were expose to hypoxia rather than just the foetuses, the physiological characteristics of the ewes should be presented (i.e. body weight and haemoglobin level). This is an important outcome for the current study, despite the model having been published elsewhere.

Why was $Q < 0.15$ chose as a threshold for significance? Generally, $Q < 0.05$ is used in the literature. This should be explained and referenced in discussion.

Minor comments

Add reference to the experimental model in the title, i.e. ovine/sheep

Referee #2:

The manuscript by McGuckin et al., aims to better understand how chronic hypoxia impacts the developing fetal liver using a sheep model of maternal hypoxia in late gestation. There is a robust number of prior publications proving the experimental model, and this manuscript further builds upon this knowledge. Overall, the manuscript is well written and there is a significant amount of data which is presented in an easy-to-understand manner. People who are interested in this area of research will find this manuscript useful and meaningful. Here are my specific comments:

1. Line 111 - how do you know this model does not result in reduced placental nutrient supply? I cannot find from the referenced literature, and it is not shown in this paper, analysis of placental nutrient transport or nutrient concentration in fetal circulation. I also think the authors need to be careful stating that this is a model "without placental insufficiency". Many would define placental insufficiency as "the failure of the placental to deliver adequate oxygen and nutrients to maintain fetal growth rates", and reduced oxygen delivery via the placenta is the exact model employed in this study. If the authors can confirm that nutrient transport across the placenta is not impacted by their maternal hypoxia model, then it would be better to state in Line 103 - "The objective of this study was to delineate how chronic hypoxia, without reduced placental nutrient supply, disrupts metabolism..."

2. Line 268, Fetal sex as a biological variable - Please acknowledge that "previous studies showing no sex differences" are in sheep. Also, given you have delineated between male and female fetuses in the graphical representation of the data, it would be nice to also see males and females separated in Table 1, even if there are no statistics.

END OF COMMENTS

To Journal of Physiology Editors and Reviewers,

We appreciate the positive feedback from the reviewers. We have considered their comments and revised our manuscript accordingly. We have also made additional revisions addressing the other editorial and formatting concerns. Our responses are listed point by point (**shown in red text below**), along with reference to the page numbers within the red-line manuscript file. We look forward to the review and acceptance of our revised manuscript.

Sincerely,

Stephanie Wesolowski

Response to Editorial Concerns

REQUIRED ITEMS

- Author photo and profile. First or joint first authors are asked to provide a short biography (no more than 100 words for one author or 150 words in total for joint first authors) and a portrait photograph. These should be uploaded and clearly labelled together in a Word document with the revised version of the manuscript. See Information for Authors for further details.

The author profile and portrait have been added to the revised manuscript.

- You must start the Methods section with a paragraph headed Ethical approval (https://jp.msubmit.net/cgi-bin/main.plex?form_type=display_requirements#methods).

Ethical approval has been moved from the Methods paragraph titled "Ovine model of hypoxic pregnancy" to the first paragraph titled "Ethical approval".

Research must comply with The Journal's policies regarding animal experiments (<https://physoc.onlinelibrary.wiley.com/hub/animal-experiments>) and adherence to these policies must be stated in the manuscript.

This is stated in the "Ethical approval" section.

Authors should confirm in their Methods section that their experiments were carried out according to the guidelines laid down by their institution's animal welfare committee, including an ethics approval reference number. The Methods section must contain a statement about access to food, water and housing, details of the anaesthetic regime: anaesthetic used, dose and route of administration, and method of killing the experimental animals.

The ethics Home Office Project Licence number PP6755721 was added to the methods. The original methods contains a statement about access to food, water, housing in the paragraph that starts on Line 126.

A statement about anaesthetic is listed in Line 139.

- Please upload separate high-quality figure files via the submission form.

We have uploaded our ppt file for the figures. In this file, images are high resolution and not compressed. If you need an additional format, please contact us.

- You must upload original, uncropped western blot/gel images (including controls) if they are not included in the manuscript. This is to confirm that no inappropriate, unethical or misleading image manipulation has occurred. These should be uploaded as 'Supporting information for review process only'. Please label/highlight the original gels so that we can clearly see which sections/lanes have been used in the manuscript figures. For more information, see: <https://physoc.onlinelibrary.wiley.com/hub/journal-policies#imagmanip>.

Images of original, uncropped blots have been provided.

- Your paper contains Supporting Information of a type that we no longer publish, including supplementary tables and figures. Any information essential to an understanding of the paper must be included as part of the main manuscript and figures. The only Supporting Information that we publish are video and audio, 3D structures, program codes and large data files. Your revised paper will be returned to you if it does not adhere to our Supporting Information Guidelines.

We have contacted the Office with questions about this. We have been permitted to include our large tables as supplementary material. We have included our larger correlation matrix as Figure A1, an appendix at the end.

- Please include an Abstract Figure file, as well as the Figure Legend text within the main article file. The Abstract Figure is a piece of artwork designed to give readers an immediate understanding of the research and should summarise the main conclusions. If possible, the image should be easily 'readable' from left to right or top to bottom. It should show the physiological relevance of the manuscript so readers can assess the importance and content of its findings. Abstract Figures should not merely recapitulate other figures in the manuscript. Please try to keep the diagram as simple as possible and without superfluous information that may distract from the main conclusion(s). Abstract Figures must be provided by authors no later than the revised manuscript stage and should be uploaded as a separate file during online submission labelled as File Type 'Abstract Figure'. Please also ensure that you include the figure legend in the main article file. All Abstract Figures should be created using BioRender. Authors should use The Journal's premium BioRender account to export high-resolution images. Details on how to use and access the premium account are included as part of this email.

The original graphical abstract has been replaced by a version exported from BioRender Premium.

EDITOR COMMENTS

Reviewing Editor:

Thank you for submitting your manuscript titled "Hypoxic Pregnancy Promotes Fibrosis and Increases Stress Metabolites in the Fetal Liver" to The Journal of Physiology. It has been assessed by an academic reviewing editor and two external peer reviewers, all of whom believe this is a very nice set of studies and a manuscript that is likely to have an influence on the field. Both reviewers have several specific comments that should be addressed in order to revise and further improve the manuscript. Please also see 'Required Items' above.

REFeree COMMENTS

Referee #1:

McGuckin et al. have demonstrated that a hypoxic for pregnant ewes leads to increased foetal hepatic collagen deposition, indicating early liver injury. They observed disruptions in central carbon metabolism, mitochondrial dysfunction, and enhanced gluconeogenesis, while mitochondrial oxidation and antioxidant capacity remained largely intact. I have detailed the strength of the manuscript below. I also have some comments that should be addressed before I can recommend for publication.

Strengths

Timely and important topic.

Comprehensive analysis of samples using advanced techniques.

Mammalian model with more similarities to human physiology than rodent models.

Robust outcomes from an experimental approach that requires high degree of expertise.

Clearly written and robust experimental methods

Figures are clear and informative.

Discussion references relevant literature.

Conclusion is valid and justified by experimental data.

Major comments

I have a major comment on the chosen model: Although I applaud the use of an ovine model, in most cases of foetal chronic hypoxia in humans, the mother remains asymptomatic because her oxygen levels and systemic circulation are sufficient to maintain her own oxygenation, even if oxygen transfer to the foetus is compromised.

Whereas in this current model, the pregnant ewes were exposed to environmental hypoxia. This model would arguably be more relevant for humans living at altitude, rather than foetal chronic hypoxia. This limits the relevance of the model to foetal chronic hypoxia in humans and this limitation should clearly be stated in the discussion.

We appreciate this concern. While our model produces fetal hypoxia, this is concomitant with maternal hypoxia. We have added text in the Discussion to address these limitations (lines 534-539).

Given that the ewes were exposed to hypoxia rather than just the foetuses, the physiological characteristics of the ewes should be presented (i.e. body weight and haemoglobin level). This is an important outcome for the current study, despite the model having been published elsewhere.

Maternal characteristics have been added to Table 1.

Why was $Q < 0.15$ chosen as a threshold for significance? Generally, $Q < 0.05$ is used in the literature. This should be explained and referenced in discussion.

Line 195 – An FDR-adjusted p-value, termed Q-value, at < 0.15 is generally accepted for discovery-based experiments. This was discussed and agreed upon with our bioinformatician, Ken Jones, co-authored on this paper. We have added text about this in our Discussion (lines 540-543).

Minor comments

Add reference to the experimental model in the title, i.e. ovine/sheep
"Ovine" has been added to the title.

Referee #2:

The manuscript by McGuckin et al., aims to better understand how chronic hypoxia impacts the developing fetal liver using a sheep model of maternal hypoxia in late gestation. There is a robust number of prior publications proving the experimental model, and this manuscript further builds upon this knowledge. Overall, the manuscript is well written and there is a significant amount of data which is presented in an easy-to-understand manner. People who are interested in this area of research will find this manuscript useful and meaningful. Here are my specific comments:

1. Line 111 - how do you know this model does not result in reduced placental nutrient supply? I cannot find from the referenced literature, and it is not shown in this paper, analysis of placental nutrient transport or nutrient concentration in fetal circulation. I also think the authors need to be careful stating that this is a model "without placental insufficiency". Many would define placental insufficiency as "the failure of the placental to deliver adequate oxygen and nutrients to maintain fetal growth rates", and reduced oxygen delivery via the placenta is the exact model employed in this study. If the authors can confirm that nutrient transport across the placenta is not impacted by their

maternal hypoxia model, then it would be better to state in Line 103 - "The objective of this study was to delineate how chronic hypoxia, without reduced placental nutrient supply, disrupts metabolism..."

In this model, placental nutrient flux has not been measured. We agree that there could be placental transport effects. Thus, we have removed "without placental insufficiency" from line 89. In lines 542-543, we have added text stating that the effects on placental function are not known and warrant future study.

2. Line 268, Fetal sex as a biological variable - Please acknowledge that "previous studies showing no sex differences" are in sheep. Also, given you have delineated between male and female fetuses in the graphical representation of the data, it would be nice to also see males and females separated in Table 1, even if there are no statistics.

We added "in fetal sheep" to the methods in the referred to section.

We feel that it would be misleading to report both the average and SD for males and females separately when the statistics were performed using the group average. If the editor finds it appropriate, we could include the separated male and female averages in a supplemental table or appendix.

Dear Dr Wesolowski,

Re: JP-RP-2025-288724R1 "**Hypoxic Pregnancy Promotes Fibrosis and Increases Stress Metabolites in the Ovine Fetal Liver**" by Molly M McGuckin, Dong Wang, Jasmine Ortiz, Evgenia Dobrinskikh, Wen Tong, Kimberley J Botting-Lawford, Youguo Niu, Dino A Giussani, and Stephanie R Wesolowski

We are pleased to tell you that your paper has been accepted for publication in The Journal of Physiology.

Yours sincerely,

Laura Bennet
Senior Editor
The Journal of Physiology

If you would like to receive our 'Research Roundup', a monthly newsletter highlighting the cutting-edge research published in The Physiological Society's family of journals (The Journal of Physiology, Experimental Physiology, Physiological Reports, The Journal of Nutritional Physiology and The Journal of Precision Medicine: Health and Disease), please click this link, fill in your name and email address and select 'Research Roundup':
<https://www.physoc.org/journals-and-media/membernews>

- You can help your research get the attention it deserves! Check out Wiley's free Promotion Guide for best-practice recommendations for promoting your work at: www.wileyauthors.com/eeo/guide. You can learn more about Wiley Editing Services which offers professional video, design, and writing services to create shareable video abstracts, infographics, conference posters, lay summaries, and research news stories for your research at: www.wileyauthors.com/eeo/promotion.

EDITOR COMMENTS

Reviewing Editor:

Thank you very much for submitting your revised manuscript to The Journal of Physiology. Reviewers and editors find it much improved and appreciate the revisions made.

REFEREE COMMENTS

Referee #1:

I thank the authors for their careful consideration of my comments and I am now able to recommend this manuscript for publication, congratulations!

Referee #2:

I am satisfied with the changes and have no further comments.